Measures of skin conductance and heart rate in alcoholic men and women during memory performance

Sawyer Kayle S. 1 2 3 kslays@bu.edu
Poey Alan 1 2 3
Ruiz Susan Mosher 1 2 3
Marinkovic Ksenija 4 5
Oscar-Berman Marlene 1 2 3
1 Boston University School of Medicine , Boston, MA , USA
2 VA Boston Healthcare System , Boston, MA , USA
3 Athinoula A. Martinos Center for Biomedical Imaging, Massachusetts General Hospital , Boston, MA , USA
4 University of California at San Diego, CA , USA
5 Psychology Department, San Diego State University, CA , USA
Abdullah Jafri
Electronic publication date: 2015 May 5
Publication date: 2015
Volume: 3
Electronic Location ID: e941
Received 2014 May 22; Accepted 2015 Apr 16
Copyright: © 2015 Sawyer et al.
Copyright year: 2015
Copyright holder: Sawyer et al.
License: This is an open access article distributed under the terms of the Creative Commons Attribution License, which permits unrestricted use, distribution, reproduction and adaptation in any medium and for any purpose provided that it is properly attributed. For attribution, the original author(s), title, publication source (PeerJ) and either DOI or URL of the article must be cited.
License URL: https://creativecommons.org/licenses/by/4.0/

Keywords: Alcoholism, Heart rate, Skin conductance, Psychophysiology, Emotion, Memory

Funding: NIAAA R01AA07112 K05AA00219 K01AA13402 R01AA016624 NCRR 1 UL1 RR025758-04 US Department of Veterans Affairs This research was supported by funds from the US Department of Health and Human Services, National Institute on Alcohol Abuse and Alcoholism (R01AA07112, K05AA00219, K01AA13402, R01AA016624) and National Center for Research Resources (1 UL1 RR025758-04, Harvard Clinical and Translational Science Center), and the Medical Research Service of the US Department of Veterans Affairs. The funders had no role in study design, data collection and analysis, decision to publish, or preparation of the manuscript.

==============================
We examined abnormalities in physiological responses to emotional stimuli associated with long-term chronic alcoholism. Skin conductance responses (SCR) and heart rate (HR) responses were measured in 32 abstinent alcoholic (ALC) and 30 healthy nonalcoholic (NC) men and women undergoing an emotional memory task in an MRI scanner. The task required participants to remember the identity of two emotionally-valenced faces presented at the onset of each trial during functional magnetic resonance imaging (fMRI) scanning. After viewing the faces, participants saw a distractor image (an alcoholic beverage, nonalcoholic beverage, or scrambled image) followed by a single probe face. The task was to decide whether the probe face matched one of the two encoded faces. Skin conductance measurements (before and after the encoded faces, distractor, and probe) were obtained from electrodes on the index and middle fingers on the left hand. HR measurements (beats per minute before and after the encoded faces, distractor, and probe) were obtained by a pulse oximeter placed on the little finger on the left hand. We expected that, relative to NC participants, the ALC participants would show reduced SCR and HR responses to the face stimuli, and that we would identify greater reactivity to the alcoholic beverage stimuli than to the distractor stimuli unrelated to alcohol. While the beverage type did not differentiate the groups, the ALC group did have reduced skin conductance and HR responses to elements of the task, as compared to the NC group.

Introduction

Abstinent alcoholics are impaired on many cognitive and emotional tasks (Oscar-Berman et al., 2014). Most investigations into these phenomena have focused on deficits reflective of abnormalities of the central nervous system (Makris et al., 2008) and of overt behaviors (Oscar-Berman et al., 2014), with less attention devoted to dysregulation of the autonomic nervous system (ANS) (Karpyak et al., 2014; Boschloo et al., 2011). Despite a certain degree of autonomy of the ANS in its control over the internal physiological milieu, central and autonomic components of the nervous system work in synchrony and interact in reciprocal ways on numerous levels of the neuraxis, regulating behavior in a seamlessly integrated manner (Clore & Ortony, 2000; Critchley, Eccles & Garfinkel, 2013; Damasio, 1998). Nevertheless, their functional profiles and measurement methods are distinct. Peripheral autonomic measures may help discern effects of chronic alcoholism on the neurofunctional systems that comprise emotional dimensions. They also may provide insight into brain modulatory processes such as arousal circuits that moderate cognition, including working memory (Dolcos & McCarthy, 2006). In the present study, we examined the peripheral responses of electrodermal activity and heart rate (HR), two commonly used indicators of ANS reactivity, in alcoholic (ALC) and nonalcoholic control (NC) individuals while they viewed emotionally evocative visual stimuli.

Electrodermal activity reflects skin conductivity (tonic, slow changes over time) and skin conductance responses (SCRs; short-lasting phasic changes in response to discrete stimuli). The conductivity of the skin is primarily influenced by the amount of moisture produced by eccrine sweat glands under control of the sympathetic branch of the ANS and is therefore an excellent indicator of the overall arousal state (the level of skin conductance) or stimulus-induced arousal (SCR) (Boucsein, 2012; Dawson, Schell & Filion, 2007; Naqvi & Bechara, 2006). The HR measure (i.e., the number of beats per minute) is controlled by both the sympathetic and parasympathetic branches of the ANS (Berntson, Quigley & Lozano, 2007), and it reflects stress responsiveness in psychosocial and cognitive situations (Karpyak et al., 2014; Starcke et al., 2013). These measures have been the methods of choice in numerous studies investigating functions that rely on affective dimensions including lie detection, orienting to novelty, classical conditioning, and processing of stimuli varying in emotional valence and significance (Dawson, Schell & Filion, 2007; Maltzman, 1990; Lang et al., 1993; MacLaren, 2001), as well as decision making (Naqvi & Bechara, 2006). Furthermore, these peripheral indices provide insight into psychopathological conditions such as schizophrenia (Schell et al., 2005), psychopathy (Fowles, 1993), and depression (Thorell, 2009). In the case of alcoholism, these indices complement behavioral or neuroimaging measures that do not address ANS reactivity. Indeed, long-term alcohol abuse results in diverse abnormalities of ANS functions. For example, reduced sweating is observed in chronic alcoholics diagnosed with peripheral neuropathy (Low et al., 1975), and post mortem studies indicate degeneration of the sympathetic and vagus nerves (Novak & Victor, 1974).

Neuroanatomical substrates that influence electrodermal and cardiac response systems consist of distributed and interrelated limbic and cortical circuitries. Direct electrical stimulation of limbic structures, primarily the amygdala, hippocampus, and anterior cingulate gyrus, have been shown to evoke large SCRs, whereas stimulation of cortical areas such as frontal and mid-temporal regions have been shown to result in a weaker, but still observable modulation (Mangina & Beuzeron-Mangina, 1996). Neuropsychological research has shown that lesions of the ventromedial prefrontal, anterior cingulate, and right parietal areas result in reduced electrodermal activity (Critchley, 2002; Tranel, 2000). Studies using simultaneous measures of central (fMRI), and autonomic activity (HR) have provided insight into the control of HR. Activity in the amygdala, insula, anterior cingulate, and brainstem predict changes in HR in response to emotional faces (Critchley, 2005).

Levels of physiological arousal have been positively associated with level of dependence and consumption in responses to alcohol cue exposure (Sinha et al., 2009; Carter & Tiffany, 1999), and gender differences in these responses also have been reported (Nesic & Duka, 2006), although not in relation to alcoholism. Chronic alcoholism also may be differently associated with HR cue reactivity. That is, ALC individuals have shown attenuated HR responses to alcohol cues in a Stroop paradigm (Stormark et al., 2000), and their HR responses to alcohol cues have been shown to differentiate beverage preferences (Staiger & White, 1991). However, when compiling a meta-analysis of 11 HR and 9 SCR cue-reactivity studies, Carter & Tiffany (1999) found that alcoholics did have higher reactivity to alcohol-related cues than to neutral cues. Of note, the perception of alcohol-related cues involves limbic structures important for emotional functions. These structures also are used in the processing of emotional faces, which has been observed to be impaired in alcoholics (Marinkovic et al., 2009).

The present study examined the intensity of ANS responses to two types of stimuli with differing emotional characteristics (facial expressions and beverage cues), and how they interacted. The stimuli were presented in the context of a memory task that was designed to investigate how emotional memory for faces would be influenced by subsequent distraction by alcohol cues, and whether the reactivity to those alcohol cues would be differentially modulated by positive or negative facial expressions (Ruiz, 2012). In addition to examining the influence of alcoholism on ANS responses, we also sought to explore if the effects we observed were different for men and women, and if those gender effects differed for the ALC and NC groups. In accordance with the literature described, we predicted that ALC participants would have abnormal SCR and HR responsivity to the facial and beverage cues presented in our task, and that men and women would display divergent ANS patterns. Specifically, we predicted that relative to NC participants, the ALC participants would have lower SCR and HR responses to the face stimuli, and greater reactivity to the alcoholic beverage stimuli than to the stimuli unrelated to alcohol. We also predicted that women would show greater ANS responses to the faces than men, in accordance with gender differences typically reported in psychophysiological responses to emotional stimuli (e.g., Bianchin & Angrilli, 2012), and we sought to explore how this gender difference would be represented in ALC participants.

In both acute and chronic alcohol exposure, orienting responses can be abnormal. Marinkovic and colleagues (2001) found that small amounts of alcohol (about one or two drinks) in healthy controls attenuated electrical activity of the brain (as measured by the event related brain potential of the orienting response) to novel sounds on trials with corresponding autonomic arousal (i.e., SCR). This indicates that the brain may be less sensitive to novel or rarely occurring stimuli under the influence of alcohol, and that the SCR is a concomitant indicator of this orienting response. Croissant and colleagues (2006) observed that acute alcohol administration attenuated the HR increases observed in response to aversive sounds and reward stimuli, but the effects depended upon gender and family history of alcoholism. Ray and colleagues (2006) reported that heavy drinkers with heightened HR reactivity to intravenous alcohol infusion were more sensitive to the invigorating properties of alcohol, while being less sensitive to the sedative and unpleasant effects of alcohol intoxication. Interestingly, we showed that alcoholic Korsakoff patients (with limbic and prefrontal damage) had low electrodermal orienting responses to unexpected loud sounds (Oscar-Berman & Gade, 1979). While orienting responses typically are associated with a reduction in HR (Weisbard & Graham, 1971), in the present study we predicted that these responses would be abnormal in ALC participants.

In summary, essential involvement of limbic and prefrontal networks in the regulation and modulation of electrodermal and cardiovascular activity makes these measures valuable for studying emotional functioning as affected by alcoholism. Therefore, we measured SCR and HR in ALC and NC participants undergoing an emotional memory task. Because a number of studies have found increased neural reactivity when ALCs are exposed to alcohol cues (Wrase et al., 2002; Myrick et al., 2004), our task employed alcohol distractor cues during a memory maintenance period for faces with emotional expressions. We predicted that ALCs would have abnormal physiological responses to the two different emotional characteristics (facial expressions and beverage cues), and that these responses would be different for men and women. Finally, we sought to explore (a) how face valence and distractor type would interact, and (b) how the memory probe faces would differentiate ANS responsivitity for ALC men and women.

Materials and Methods

Participants

A total of 62 individuals from the Boston area were included in the study (see Table 1). One group consisted of 32 abstinent ALC men and women, and the other group consisted of 30 gender- and age-equivalent healthy NC individuals. The task in which they participated took place in an MRI scanner, where we collected neuroimaging and behavioral data concurrently with psychophysiological measures. (The neuroimaging and behavioral results are described separately by Ruiz, 2012.) Originally, SCR and HR data were collected from 76 participants. However, because of problems encountered during physiological recording in the MRI scanner (e.g., MRI radio frequency interference, motion induced artifacts from the static magnetic field, poor skin responsivity with electrodes, other motion artifacts, etc.), we had good quality data with low levels of noise and easily discernable SCRs for 62 participants. From these 62 participants, data for a total of 46 individuals were available for analyses of each of the two physiological measures (see Tables S1 and S2 for the distribution of participants into SCR or HR subgroups, and see Supplemental Raw Physiological Data at Figshare: http://dx.doi.org/10.6084/m9.figshare.1025792). Participants responded to flyers posted online, at the VA Boston Healthcare System and the Boston University Medical Center, and in public places (churches, stores, etc.). This research was approved by the Institutional Review Boards of Boston University School of Medicine (#H24686), VA Boston Healthcare System (#1017 and #1018), and Massachusetts General Hospital (#2000P001891). Participants were reimbursed for time and travel expenses.

Table 1 Characteristics of the research participants.

Means and standard deviations (SD) are displayed for age, education, measures of drinking history, the Wechsler Adult Intelligence Scale (WAIS) and the Wechsler Memory Scale (WMS). Besides amount and duration of drinking, these characteristics did not differ significantly between the ALC and NC groups, with the exception of education, for which the NC group had more years on average (95% CI [0.0–1.7] years). Tables S1 and S2 describe the subgroups examined for heart rate and SCR. For those samples, the ALC and NC groups had similar education levels.

	Alcoholic participants	
	All (N = 32)	Women (N = 16)	Men (N = 16)	
	Mean	SD	Mean	SD	Mean	SD	
Age (years)	55.2	9.9	56.9	9.4	53.5	10.5	
Education (years)	14.9	2.0	15.7	2.0	14.1	1.8	
Duration of Heavy Drinking (years)	16.2	5.9	15.3	4.9	17.1	6.9	
Daily Drinks	11.7	9.3	10.1	8.3	13.4	10.3	
Length of Sobriety (years)	8.8	11.2	11.5	12.0	5.9	9.8	
WAIS-III Full Scale IQ	109.1	14.3	110.1	13.0	108.1	15.8	
WAIS-III Verbal IQ	111.5	12.0	112.8	11.6	110.3	12.6	
WAIS-III Performance IQ	104.1	17.4	104.9	16.4	103.3	18.8	
WMS-III Immediate Memory	109.8	17.9	113.6	20.4	105.9	14.7	
WMS-III Delayed Memory	112.0	18.5	115.6	22.9	108.4	12.3	
WMS-III Working Memory	103.6	12.2	104.8	12.4	102.4	12.4	
	Nonalcoholic participants	
	All (N = 30)	Women (N = 15)	Men (N = 15)	
	Mean	SD	Mean	SD	Mean	SD	
Age (years)	52.6	12.8	54.7	15.2	50.5	10.0	
Education (years)	15.9	2.1	16.1	2.4	15.6	1.8	
Duration of Heavy Drinking (years)a	0.1	0.5	0.0	0.0	0.3	0.7	
Daily Drinks	0.3	0.6	0.3	0.7	0.3	0.5	
Length of Sobriety (years)	NA	NA	NA	NA	NA	NA	
WAIS-III Full Scale IQ	111.5	13.9	114.0	17.2	109.2	10.0	
WAIS-III Verbal IQ	113.3	15.0	114.2	17.7	112.5	12.5	
WAIS-III Performance IQ	107.4	14.5	110.8	14.5	104.3	14.2	
WMS-III Immediate Memory	111.1	16.3	115.2	17.2	107.2	14.9	
WMS-III Delayed Memory	110.5	14.1	112.2	14.5	108.9	14.1	
WMS-III Working Memory	106.3	12.7	108.6	13.5	104.2	11.9	
Notes.

a One man drank during his time in the military service for about 2.5 years, decades prior to the scan, but this drinking was not severe and he reported that it did not impact his occupation, health, or personal life. Another man reported drinking heavily for 0 years in a prior study in our lab, but then after answering the same questionnaire for the present study, he reported that he recalled that, decades prior to the scan, he drank over 21 drinks per week for approximately a year. One woman reported that at most she drank a half bottle of wine a day (1–3 drinks) for at most 5 years, so we used 5 years as a conservative estimate. In prior (and subsequent) studies in our laboratory, she reported drinking less, so her DHD was set to 0 for those studies. All three participants had not been drinking heavily recently.

Selection procedures included an initial structured telephone interview to determine age, level of education, health history, and history of alcohol and drug use. To be brought in for neuropsychological testing, electrophysiological measurements, and neuroimaging, all potential participants were required to be right-handed, have normal or corrected-to-normal vision, and speak English as their first language (or have acquired English as a second language by five years of age). Eligible individuals were invited to the laboratory for further screening and evaluations.

Participants received a structured interview regarding their drinking patterns, including length of abstinence and duration of heavy drinking (DHD), i.e., the number of years they consumed more than 21 drinks per week (one drink: 355 mL beer, 148 mL wine, or 44 mL hard liquor). A Quantity Frequency Index (QFI; Cahalan, Cisin & Crossley, 1969), which roughly corresponds to number of daily drinks (at one ounce of ethanol per drink), was calculated for each participant. This measure evaluates the amount, type, and frequency of alcohol usage over the last six months (for the NC participants), or over the six months preceding cessation of drinking (for the ALC participants). The ALC participants met DSM-IV criteria for moderate to severe lifetime alcohol abuse or dependence, and had a minimum duration of five years of heavy drinking (at 21 or more drinks per week). All ALCs had abstained from drinking alcohol for at least four weeks prior to testing (except one ALC man who had a drink three days prior to the study and had ceased heavy drinking eight years earlier).

Neurobehavioral and psychiatric evaluations typically required six to nine hours over three or more days. Participants had frequent breaks, and sessions were discontinued and rescheduled if a participant indicated fatigue. Participants underwent a medical history interview and vision testing, plus a series of questionnaires (e.g., handedness, alcohol and drug use) to ensure they met inclusion criteria. In order to minimize confounding effects from illicit drug use, psychoactive drug use, and psychiatric comorbidity, participants were given an extensive battery of screening tests. They performed a computer-assisted, shortened version of the Diagnostic Interview Schedule (DIS; Robins et al., 2000) Version IV that provides lifetime psychiatric diagnoses according to the Diagnostic and Statistical Manual criteria (DSM-IV-TR; APA, 2000). Individuals were excluded from further participation if any source (DIS scores, hospital records, referrals, or personal interviews) indicated that they had any of the following: Korsakoff’s syndrome; HIV; hepatitis; cirrhosis; major head injury with loss of consciousness greater than 20 min; stroke; epilepsy or seizures unrelated to alcoholism; Hamilton Rating Scale for Depression (Hamilton, 1960) score over 14; major depressive disorder; bipolar I or II disorder; schizoaffective disorder; schizophreniform disorder; schizophrenia; generalized anxiety disorder; or electroconvulsive therapy. Additionally, we excluded individuals who failed screening for MRI safety (e.g., metal implants, obesity, pregnancy).

Neuropsychological assessment

In order to collect demographic information, all participants were given a battery of neuropsychological tests, and the following IQ and memory measures (Wechsler, 1997) were examined: The Wechsler Adult Intelligence Scale (WAIS-III) Verbal, Performance, and Full Scale Intelligence Quotient scores (VIQ, PIQ, and FSIQ) and the Wechsler Memory Scale (WMS) Immediate, Delayed, and Working Memory scores. Independent-samples t-tests did not reveal significant differences among the ALC and NC groups for the WAIS and WMS measures.

Memory task

All participants were given a delayed memory task in the MRI scanner, in which face and alcoholic beverage stimuli were displayed. The task was designed to assess multiple cognitive functions. The primary goals were to assess ANS reactivity (SCR and HR) to emotional face stimuli and to alcohol cues. The secondary goal of the task was to examine the interaction of these factors (Face Emotion x Distractor Type) in order to characterize their combined influence on ANS reactivity during the distractor element and the memory probe (not on memory performance itself, which is addressed elsewhere; Ruiz, 2012).

Our task has been described in detail by Ruiz (2012). Figure 1 shows the flow of the task for three example trials. On each trial, pictures of two faces were displayed simultaneously for three seconds, followed by a fixation asterisk (*) for one second (the encoded faces). The participants were asked to maintain these faces in memory while a colored distractor stimulus was shown (the distractor element). Each distractor picture was shown for three seconds, followed by a fixation asterisk (*) for one second. Following the distractor picture, a single probe face was shown for two seconds (the probe element), and the participants were required to report whether this face was one of the two faces they had just seen. Each trial was 10 s in length, and was followed by a variable delay period (with a mean duration of 10 s, ranging from 2 to 22 s) during which the participants engaged in fixation on a set of crosshairs (+ + +). The delay period was designed to be of sufficient duration to assess orienting to the encoded faces that followed it, for which we predicted abnormal ANS responses in the ALC group.

Figure 1 This figure shows the flow of the delayed match-to-sample task paradigm.

Rows provide examples of the three facial emotion types (positive, neutral, negative) and the three distractor types (scrambled, nonalcoholic, alcoholic). The rows also represent both matching and non-matching probe faces (match, nonmatch, match). The faces have been obscured in this publication to protect the identity of the individuals who were photographed. The participants saw the entire un-obscured faces.

The face stimuli were divided into three valence types by facial expression: positive, negative, and neutral (the Face Valence). Photographs were of unfamiliar adults without glasses, facial hair, or jewelry. The faces were shown in grayscale against a black background. The faces were balanced to contain 50% male and 50% female faces. The photographs were evaluated for the consistency of emotional expressions by a group of independent judges, and we selected their most consistent choices (Marinkovic & Halgren, 1998). We predicted that because the brain structures responsible for ANS responses to emotional faces are dysregulated in alcoholics, the ALC group would have attenuated ANS responses to the emotional faces.

On different trials, the distractor stimulus was either a picture of an alcoholic beverage (beer, wine, liquor, or mixed drink), a picture of a nonalcoholic beverage (water, juice, milk, soda, coffee, tea, etc.), or a scrambled nonsense picture constructed to have similar colors and brightness as the other distractors (the Distractor Type). Alcoholic and nonalcoholic beverage pictures were a combination of images used with permission from the Normative Appetitive Picture System (NAPS) (Stritzke et al., 2004), and other previously published works on alcohol cues (Wrase et al., 2002; Myrick et al., 2004). Additional distractor images were modified from digital photographs taken at bars, liquor stores, and convenience stores. The scrambled images were created by inverting half the alcoholic and half the nonalcoholic beverage images and distorting them until they were not recognizable as objects. In accordance with our hypotheses, we expected that the ALC group would have stronger neuropsychological responses to the alcohol cues than the nonalcohol cues, and that this difference would be reflected by abnormal ANS responses. Further, we predicted this difference would be more pronounced for the ALC group than the NC group.

The probe face matched one of the encoded faces on 50% of the trials, and match/mismatch trials appeared in a randomized order within each run (the Probe Match). Responses were made by pressing one of two buttons with the index finger (match) or middle finger (mismatch) of the right hand. Participants were instructed to respond as quickly as possible without sacrificing accuracy. Additionally, they were told that if they felt their first response was incorrect, they could immediately correct the response by pressing the opposite button as soon as possible after the error was detected.

The task was divided into nine runs, each of which contained 18 trials. There were nine trial types made up of each combination of Face Valence, Distractor Type (e.g., positive faces followed by alcohol distractor), and Probe Match status, for a total of 18 trial types. Each emotion-distractor combination appeared twice per run. In total, there were 54 trials for each type of distractor (combined across Face Valence) and each face emotion (combined across Distractor Type), for a total of 162 trials across the entire scan. A total of 162 different faces (balanced by gender) were used in the study. Within a trial, the two encoded faces and the probe face had the same emotional expression and gender. This way, on match trials the probe facial image was identical to one of the encoded images, and on mismatch trials the facial identity changed but the emotional expression and gender did not.

An IBM ThinkPad running Presentation version 11.2 for Windows XP (NeuroBehavioral Systems, Albany, California, USA) software was used for visual presentation of the experimental stimuli and collection of participants’ responses in the scanner. Equipment for the task consisted of a shielded projector and screen, mirrors, and button box. Stimuli were back-projected onto a screen at the back of the scanner bore and were viewed by the participants through a mirror mounted to the head coil. All participants wore earplugs to attenuate scanner noise.

Psychophysiological measures

Psychophysiological data for skin conductance and pulse oximetry (for HR) were collected with MRI-compatible equipment simultaneously with fMRI data collection during the memory task. Tonic levels and phasic changes of skin conductance were recorded from MLT117F silver–silver chloride contoured finger electrodes filled with BioPac GEL101 isotonic electrode gel. Electrodes were attached to the index and middle fingers of the left hand with velcro straps, leaving the right hand free for performing the task. Skin conductance was amplified using an ADInstruments PowerLab 16Sp (ADInstruments, Dunedin, New Zealand) with ML116 GSR Amplifier, and recorded digitally using ADInstruments LabChart software. It was recorded continuously throughout the functional scans, which allowed us to gather electrodermal responses elicited during the task. Preprocessing included attenuation of noise elements via individualized notch filters for signal induced from respiratory and pulsatile movements within the magnetic field by targeting the frequencies obtained from simultaneous etCO2 measurement (approximately 0.3 Hz) and from pulse oximetry determined HR (at approximately 1.2 Hz). Fourier transformed power spectra were examined for other sources of noise, and further notch filters were applied as needed.

HR was obtained through a pulse oximeter attached to the little finger on the left hand. HR was amplified by an Invivo Magnitude 3155A system (Invivo Therapeutics, Cambridge, Massachusetts, USA), relayed to the ADInstruments PowerLab. LabChart software was utilized to identify the peak slope, and pulse identification was manually examined across each run for each subject. HR was calculated using the inter-beat interval and smoothed with a one second window. To account for signal dropouts, a 50 s moving window average was calculated, and any inter-beat intervals deviating by more than 25 BMP were replaced with the moving window average. We did not correct for respiratory sinus arrhythmia.

Statistical analyses

SPSS version 20 (IBM, Armonk, New York, USA) was used for statistical analyses, and Excel 2011 was used for graphing (Microsoft, Redmond, Washington, USA). For each element of the memory task (encoded faces, distractor element, and probe element), skin conductance was analyzed for amplitude increases. Specifically, the SCR amplitude obtained for each stimulus was calculated by measuring the level of increase from (a) the mean conductance level obtained during the three seconds preceding the element, to (b) the peak conductance during the six seconds following the element. In other words, the SCR for each trial was defined as the amount of conductance increase for that element (Dawson, Schell & Filion, 2007). This was accomplished automatically with LabChart software, without the experimenters’ input. Then, in an effort to minimize the effects of occasional outliers that were due to movement or other types of artifacts, we detected individual trial SCRs that were greater than four standard deviations above the mean (approximately 1% of the trials), and removed them from the analyses.

We assessed the skin conductance responsivity in two ways that allowed us to obtain the average, i.e., representative, values for the factors of interest without individual trial variability. First, in order to get insight into the overall level of responsivity, we obtained the counts of responses that were greater than 0.03 µS irrespective of their amplitude. Additionally, we analyzed response amplitudes by averaging them across all runs for each trial type irrespective of the response frequency. Models were constructed for the responses to the encoded faces, distractor element, and probe element separately. Between-subjects interaction effects were included to test the influence of preceding stimuli. The data were analyzed with full-factorial mixed-design analyses of variance (ANOVA) that included: (a) between-group factors of Group and Gender, and (b) the relevant within-subject factors of Face Valence, Distractor Type, and Probe Match, and all interactions of these factors (Woodward, Bonett & Brecht, 1990).

As with skin conductance analyses, HR changes were examined for each element of the task. Difference values were obtained by subtracting the mean HR in the two seconds following the element from the two seconds preceding it. As with the SCR analyses, outliers were detected and removed, defined as any single-trial HR changes greater than five beats per minute (BPM) or exceeding four standard deviations. Around 1% of trials were removed in this manner. As with the SCR analyses, the HR data were averaged across the stimulus categories included in each ANOVA design, which used Group and Gender as between-group factors, and Face Valence, Distractor Type, and Probe Match as within-subject factors, along with all interactions of these factors.

For both SCR and HR analyses, the full-factorial mixed-design ANOVA models simultaneously provided results not only for the specific predictions to our hypotheses, but also for further exploratory analyses. That is, the ANOVA models for the encoded faces tested our predictions that the orienting responses would be reduced for the ALC group, and that the responses to the emotional faces would be diminished, while additionally including other exploratory effects specified in the full-factorial model. For the distractor element, the ANOVA models included factors to examine our prediction that the alcohol cues would elicit stronger ANS responses for the ALC group, and the Group × Distractor Type interaction would investigate whether this difference were more pronounced for the ALC group than for the NC group. Further analyses, including an examination of how the face valence differently influenced responses to each distractor type (Face Valence × Distractor Type interaction), were conducted on an exploratory basis. Interactions with Gender were included in the models to address our hypothesis that ANS reactions would be stronger for the women than for the men, and to explore how this difference is represented among alcoholics.

Results

The results of the SCR and HR measures were in line with our predictions that relative to NC participants, the ALC participants would show reduced SCR and HR responses to the face stimuli. We did not find evidence to support our hypotheses that alcoholics would have greater reactivity to the alcoholic beverage stimuli than to the distractor stimuli unrelated to alcohol, nor that women would demonstrate greater ANS reactivity than men.

Skin conductance

Both types of skin conductance measures, i.e., response counts and response amplitudes, were analyzed for each of the task elements (encoded faces, distractor element, and probe element). In concert with our predictions, ALC participants manifested lower skin conductance responsivity than the NCs, and this was particularly apparent for the ALC women. In addition to Group and Gender, SCR was sensitive to emotional face expressions and the matching status of the probe element. Results of the analyses for each of the three task elements are presented seriatim.

Encoded faces

A mixed-design ANOVA was used that included the following factors: Group, Gender, and Face Valence, averaging across the other conditions before statistical analyses. For the SCR counts, the significant main effect of Group (F (1, 41) = 7.9, p < 0.01) indicated that the ALC participants produced 10% points fewer responses above 0.03 µS than the NCs, as predicted. This effect was particularly evident for female participants (F (1, 41) = 5.5, p < 0.05; Fig. 2), although the Group × Gender interaction was not significant (F (1, 41) = 0.9, p = 0.34). Analyses of the SCR amplitudes did not reveal significant results for our hypotheses that the encoded faces would elicit a diminished orienting response in alcoholics (F (1, 41) = 1.8, p = 0.18), nor abnormal responses to the emotional faces as indicated by a Group × Face Valence interaction (F (2, 82) = 0.5, p = 0.61). Exploratory analyses did not yield significant results.

Figure 2 For the encoded faces, alcoholic participants had fewer skin conductance responses (SCR) than nonalcoholic participants.

Data are presented for alcoholic participants (ALC, in red) and nonalcoholic participants (NC, in blue), split by gender. The asterisk indicates a significant difference, and error bars represent standard error. No significant interactions were identified. (A) ALC women had significantly fewer SCRs (>0.03 microsiemens (µS)) to the encoded faces than ALC men. (B) SCR amplitudes (µS) to the same stimuli appeared to be similar.

Distractor element

A mixed-design ANOVA was used that included the following factors: Group, Gender, Face Valence, and Distractor Type, averaging across the other conditions before the analyses. As was the case for encoded faces, the ALC participants were significantly less responsive than the NCs to the distractors, as reflected in 13% points fewer measurable SCRs (F (1, 41) = 11.3, p < 0.01). The ALC women were significantly less responsive than the NC women (F (1, 41) = 8.2, p < 0.001). A more detailed, but consistent picture emerged when the SCR amplitudes were analyzed (Fig. 3), although the predicted Group × Distractor Type interaction was not identified (F (2, 82) = 1.1, p = 0.34; Fig. 4). The main effect of Group (F (1, 41) = 2.8, p = 0.1) was especially evident for the difference between the female ALCs and NCs (F (1, 41) = 4.2, p < 0.05), though the interaction was not significant (F (1, 41) = 1.7, p = 0.20), nor was the exploratory analysis of Face Valence × Distractor Type (F (4, 164) = 0.9, p = 0.45).

Figure 3 Skin conductance responses (SCR) to distractor cues were reduced for alcoholic participants compared to nonalcoholic controls.

Data are presented for alcoholic participants (ALC, in red) and nonalcoholic participants (NC, in blue), split by gender. Asterisks indicate significant differences, and error bars represent standard error. No significant interactions were identified. (A) ALC women had significantly fewer SCRs (>0.03 microsiemens (µS)) to the distractor images than ALC men. (B) SCR amplitudes (µS) to the same stimuli revealed a similar pattern.

Figure 4 Skin conductance responses (SCR) to the distractor cues were not significantly related to the type of distractor for alcoholic participants compared to nonalcoholic controls.

Data are presented for alcoholic participants (ALC, in red) and nonalcoholic participants (NC, in blue). Error bars represent standard error. (A) A significant interaction of Group × Distractor Type was not identified for SCR counts. (B) SCR amplitudes (µS) also did not reveal a significant interaction.

Probe element

A mixed-design ANOVA included the following factors: Group, Gender, Face Valence, Distractor Type, and Probe Match (averaging across runs), and was applied to explore effects for both types of skin conductance measures. The significant main effect of Group (F (1, 41) = 6.6, p < 0.05) for the SCR counts once again confirmed that the ALC group produced 14% points fewer SCRs than the NC group, and this effect was particularly prominent for female participants (F (1, 41) = 4.1, p < 0.05). The main effect of Probe Match (F (1, 41) = 8.7, p < 0.01), was a result of greater numbers of responses to matching probe stimuli. A significant Gender × Probe Match interaction (F (1, 41) = 6.0, p < 0.05) indicated that male participants were especially sensitive to the match/mismatch difference (F (1, 41) = 15.3, p < 0.01; Fig. 5).

Figure 5 Skin conductance responses (SCR) to probe faces were reduced for alcoholic participants compared to nonalcoholic controls.

Nonmatch SCRs are in blue and match SCRs are in red. Asterisks indicate significant differences between match status, and error bars represent standard error. No significant interactions were identified. (A) Compared to women, men had significantly more SCRs (>0.03 microsiemens (µS)) to the matching probe faces than the nonmatching probe faces. The data are shown for all alcoholic and nonalcoholic participants together. (B) For neutral and positive faces, SCR amplitudes were higher for faces that matched the encoded faces. Data are shown for alcoholic and nonalcoholic men and women together.

Analyses of SCR amplitudes imparted complementary information. A significant Face Valence x Probe Match interaction (F (1, 41) = 4.9, p < 0.01) provided insight into the relative influence of the Probe Match status vs. Face Valence. Faces with negative emotional expressions elicited larger SCRs than both positive and neutral faces (F (1, 41) = 5.5, p < 0.05), confirming the sensitivity of this measure to emotionally arousing stimuli. The match and nonmatch conditions did not differ significantly for negative face expressions. In contrast, matching face stimuli elicited larger SCRs when the expressions were neutral (F (1, 41) = 11.0, p < 0.01) and positive (F (1, 41) = 9.3, p < 0.01). The exploratory analysis of the Face Valence x Distractor Type interaction did not yield significant results (F (4, 164) = 0.2, p = 0.92).

Heart rate

Differences in the HR change (see Figs. 6–9) were analyzed with mixed-design ANOVAs, as were performed for SCR. While our predictions of reduced ANS reactions of the ALC group were generally born out, the measures of HR change did not appear to be as sensitive as SCR to Group and Gender, nor to task manipulations. Results of the analyses for each of the three task elements—encoded faces, distractor element, and probe element—are presented seriatim.

Figure 6 Heart rate (HR) tended to decline more in alcoholics in response to the encoded faces than in nonalcoholics.

Bar height indicates beats per minute (BPM). Data are presented for alcoholic participants (ALC, in red) and nonalcoholic participants (NC, in blue), split by gender. The asterisk indicates a significant difference, and error bars represent standard error. No significant interactions were identified.

Figure 7 The increase in heart rate (HR) to distractor cues depended upon the group and the emotion of the preceding face.

Heart rate (HR) increased by 0.3 beats per minute (BPM) in response to the distractor element, and a significant Group × Gender × Face Valence interaction was identified. Data are presented for alcoholic participants (ALC, in red) and nonalcoholic participants (NC, in blue). Error bars represent standard error. (A) Shows values for men. (B) Shows values for women.

Figure 8 Heart Rate (HR) responses to the distractor cues were not significantly related to the type of distractor for alcoholic participants compared to nonalcoholic controls.

Data are presented for alcoholic participants (ALC, in red) and nonalcoholic participants (NC, in blue). Error bars represent standard error. A significant interaction of Group × Distractor Type was not identified for HR counts.

Figure 9 Heart rate (HR) significantly declined but was not significantly related to the group, the type of distractor, or the emotion of the face presented for the probe.

Bar height indicates beats per minute (BPM), with standard error bars. Data are presented for alcoholic participants (ALC, in red) and nonalcoholic participants (NC, in blue), split by gender. No significant interactions were identified.

Encoded faces

As predicted by our hypotheses, the HR response for the ALC group was lower in comparison to the NC group (F (1,39) = 9.6, p < 0.01; Fig. 6). The Group difference observed in relation to the faces was most evident among the women: The HR of the ALC women was 0.3 BPM slower than the HR of the NC women (F (1,39) = 10.4, p < 0.01). The difference between ALC and NC men (0.1 BPM), while in the same direction as for the women, was not significant. We did not identify a significant Group × Face Valence interaction (F (2, 78) = 0.01, p = 0.98).

Distractor element

Analyses of the HRs of all participants combined indicated a 0.3 BPM overall increase in HR to the distractor element (t (42) = 4.2, p < 0.01). Figure 7 shows the results of the three-way interaction of Group × Gender × Face Valence, which approached significance (F (2,78) = 2.4, p = 0.1) as indicated by larger HR increases for ALC women to distractor cues following negative emotional expressions than HR increases for NC women for the same condition (F (1,39) = 3.4, p < 0.1). Our hypothesis that differential HR responses to the Distractor Type would differ by Group was not confirmed (F (2,78) = 0.72, p = 0.49; Fig. 8).

Probe element

Exploratory analyses of the HRs of all participants together to the probe face element revealed a significant overall HR reduction of 0.5 BPM (t(42) = − 5.5, p < 0.01; Fig. 9). Differences in HR response also were examined for the factors of Group, Gender, Face Valence, Distractor Type, and Probe Match, averaging across runs. The reduction in HR was similar for the ALC and NC groups for both men (95% confidence interval of NC minus ALC: [−0.8, 0.3] BPM) and women ([−0.8, 0.3] BPM); no significant effects relating to task conditions or interactions with Gender were observed for the probe element.

Discussion

In the present study, the pattern of autonomic responses to all three elements of the task (encoded faces, distractor, and probe) differed for the ALC and NC groups. The electrodermal measures indicated that, compared to the NC group, the ALC group evidenced fewer SCRs to each of the three task elements (Figs. 2 and 3), in line with our primary hypothesis. HR responses also differed between ALC and NC groups to the task elements, most notably in relation to the encoded faces, wherein the ALC group had a greater HR decline than the NC group (Fig. 6). The ALCs had a greater HR increase than NCs to the distractor element (Fig. 7), but perhaps a more similar HR decrease in response to the probe element (Fig. 9). We did not find evidence for our secondary prediction that the ALC group would have a greater difference in SCR and HR response to the alcoholic beverage distractors than the nonalcoholic beverage distractors (Figs. 4 and 8). In our exploration of gender, we determined that the alcoholism-related SCR and HR response abnormalities were pronounced among women. Taken together, these abnormalities resulting from peripheral ANS measures can be interpreted within the context of the central nervous system.

Brain systems and ANS activity

Measures of electrodermal activity and HR reflect activity of the ANS, and they have been used in studies of emotional arousal, classical conditioning, orienting to novelty and attention, cognitive load, and reward significance (Boucsein, 2012; Andreassi, 2007; Critchley, 2005; Dawson, Schell & Filion, 2007). Autonomic responses have been measured in studies of chronic alcoholics with mixed results (Maltzman & Marinkovic, 1996). Some studies failed to find abnormalities in responsivity (Wrase et al., 2002; Reid et al., 2006; Pomerleau et al., 1983), or have observed lower sensitivity and greater habituation in alcoholic participants compared to nonalcoholic controls (Finn et al., 2001). In the present study, SCR results indicated lower sympathetic arousal levels in alcoholics, in agreement with previous reports of lower skin conductance orienting responses to novel stimuli in alcoholic Korsakoff patients (Oscar-Berman & Gade, 1979). Even though the participants in the present study did not exhibit Korsakoff symptomatology, the lower autonomic responsivity may be indicative of mesocorticolimbic impairments, as (1) lesions of the amygdala (Bechara et al., 1999) and prefrontal cortex (Bechara et al., 2005b) have been shown to result in a decrease or absence of SCRs, and (2) SCRs can be elicited by electrical stimulation of amygdala, hippocampus, anterior cingulate, and frontal cortex (Mangina & Beuzeron-Mangina, 1996).

Additionally, we have found (Marinkovic et al., 2009) that abstinent alcoholics, while viewing faces with emotional expressions, exhibited blunted fMRI activation in the amygdala and hippocampus, whereas nonalcoholic controls showed robust activation in these brain areas. Moreover, the alcoholics in that study recruited the prefrontal cortex to process facial emotions, possibly compensating for the limbic deficiency (Oscar-Berman et al., 2014). While higher order cortical association areas allow us to understand the significance of emotional situations, lower level structures, e.g., those with limbic system connections, are needed to express emotions through changes in the periphery. Frontal brain systems participate in both of these processes (reviewed by Damasio, 1998; Davidson, Jackson & Kalin, 2000; Hariri et al., 2003; Roy, Shohamy & Wager, 2012; Stuss & Knight, 2012), and patients with frontal system damage not only lack emotional propriety, they also do not show changes in peripheral responses that normally accompany emotional arousal (Bechara et al., 2005b; Gainotti, 2000).

Encoding of emotional faces

Our results indicated lower skin conductance reactivity for the ALC group than for the NC group in response to the encoded faces, as predicted by our hypotheses. This finding is consistent with reports of impaired memory for emotional facial expressions (Foisy et al., 2007) and reduced cortico-limbic system reactivity to emotional faces (Marinkovic et al., 2009; O’Daly et al., 2012; Salloum et al., 2007) observed in association with alcoholism. The specific brain regions showing blunted activation in alcoholics are the amygdala and hippocampus, as well as cingulate, orbitofrontal, and insular cortex, all regions that mediate emotional processing.

We further identified that HR fell more for the ALC group than the NC group in response to the encoded faces. Although the literature on HR in alcoholics is limited by numerous methodological differences among studies, especially with respect to subject samples, a review of 10 studies examining HR variability in alcoholics (Karpyak et al., 2014) showed several consistent findings: There was a decrease in HR variability indices among alcoholics compared to controls, possibly reflecting decreased parasympathetic influences, with the effect size being proportional to amount of alcohol use, and improvement (more variability) noted after prolonged abstinence. We expected an HR decrease for both groups to the encoded faces, because a reduction is associated with a normal, healthy orienting to a stimulus (Weisbard & Graham, 1971). Unexpectedly, we observed a larger HR decrease in the ALCs, which may suggest that the emotional or psychosocial content represented by the faces was more arousing for the ALCs. Alternatively, it is possible that the apparently opposite abnormalities observed for HR and SCR reflect the differential contributions of the sympathetic and parasympathetic ANS. That is, skin conductance is influenced primarily by the sympathetic ANS, while HR is controlled by both the sympathetic and parasympathetic ANS (Berntson, Quigley & Lozano, 2007). Thus, the combined HR and SCR results might be related to differential perturbations of the sympathetic and parasympathetic ANS in alcoholism.

Distractor element

Although the ALCs showed reduced SCRs to the distractors as a whole, there was not evidence for differential effects that were specific to the alcohol cue distractor images. While studies that have measured skin conductance in alcoholics viewing alcohol-related stimuli typically have reported increased psychophysiological reactivity (Cooney et al., 1997; Laberg et al., 1992; Szegedi et al., 2000), some did not identify differences between electrodermal responsivity of alcoholics and of controls (Grüsser et al., 2004; Wrase et al., 2002). In part, our results are consistent with elements of all these studies in that they are variable and indicate small SCR effects that are not easily distinguished. These results might suggest that the ALCs in this study, who had been abstinent for a mean duration of about nine years, may have successfully maintained their sobriety in part because they no longer experience as intense ANS responses specific to alcohol-related stimuli. While we did detect mildly muted SCR to distractor cues in ALC participants, we did not identify as clear a pattern related to HR responses.

Our task was designed with the secondary aim to examine how the preceding emotional valence of the faces would influence ANS responses to the distractor element. However, we did not identify a significant interaction, likely due to the relatively low sensitivity of these autonomic measures, and by the fact that the faces were not very effective in eliciting emotional arousal.

Probe element

With regard to the probe face, while group differences in the HR responsivity were not apparent, our exploratory analyses did identify a reduction in SCR. The probe face element of the task required the participants to make a decision as to whether the face was a match or mismatch. Because alcoholics may suffer from decision-making impairments, Bechara and colleagues employed a gambling task while measuring SCRs during task performance to investigate somatic markers of substance abuse (Bechara et al., 2005a; Bechara & Damasio, 2002). The somatic-marker hypothesis posits that decision-making is a process that depends on emotion, and that deficits in emotional signaling lead to poor decision-making. The investigators found that alcoholic subjects were impaired on the task and unable to generate anticipatory SCRs while pondering risky choices, and the authors made a connection between compulsive/uncontrollable behavior and drinking. They also noted that a subgroup of substance dependent individuals who performed poorly (opting for high immediate gains in spite of future losses) had impaired anticipatory SCRs linked to dysfunction of the ventromedial prefrontal cortex and the amygdala. However, because a group of nonalcoholic patients with ventromedial prefrontal damage could generate SCRs when they received a reward or a punishment, while patients with amygdala damage could not, the investigators suggested that the roles played by the amygdala and ventromedial prefrontal cortex in decision-making are different. Therefore, it is possible that our results, confirming our hypothesis that ALC participants would demonstrate abnormal SCRs, could be accounted for by the use of different decision-making processes compared to NC participants.

Gender differences

For all three aspects of the task wherein SCR reductions were identified among alcoholics relative to controls, this reduction was particularly pronounced in the women, suggesting a greater sensitivity among women to alcohol’s long-term effects on electrodermal responsivity. Regardless of alcoholism status, men showed a larger impact of match status on SCR responses to the probe element, with men having higher SCR responses when the probe face matched than when it was a mismatch (Fig. 5).

Gender differences in HR responses were most evident following the encoded faces, with alcoholic women showing stronger HR response reductions than control women. The general trend toward increased HR responsivity to the distractors among alcoholics was clearest in the women. By contrast, the dampened HR response to the probe faces among alcoholics was similar for both men and women.

The tendency for women to display greater effects of alcoholism on ANS responses to the task element may be reflective of gender differences typically reported in psychophysiological responses to emotional stimuli (e.g., Bianchin & Angrilli, 2012). Gender in relation to peripheral and central nervous system effects of alcoholism only recently has been the focus of intensive research, and additional studies are needed (Ammendola et al., 2000; Devaud, Alele & Ritu, 2003; Ruiz & Oscar-Berman, 2013).

Conclusions

The pattern of psychophysiological reactions we observed was abnormal for the ALC group, depending upon the type of distractor and facial expression presented. For SCR, the ALC group had consistently reduced responsivity regardless of the task element. For HR, the effect of alcoholism was dependent upon the elements of the task involved. The pattern of ANS activity to emotionally laden stimuli is complex, involving widespread brain circuitry. For both SCR and HR, the alcoholism-related findings were likely due to abnormalities in the mesocorticolimbic system that controls affective functioning (Oscar-Berman et al., 2014). Our results also supported the notion of gender differences in association with long-term chronic alcoholism. Men and women differed with respect to ANS responses, with the alcoholic women generally showing more clear-cut effects than their male counterparts.

Supplemental Information

Supplemental Information 1 Raw data spreadsheet, in comma space delimited format

Each row represents data for one research participant. The columns represent various features, including demographic attributes, the authors’ subjective assessment of data quality, and the changes in physiological measures in response to each stimulus, as described in the methods. For the physiological measures, here is an example that describes how the stimuli were coded: HR-distractor-run1t11-f-neg-alcohol-no_after: Heart rate in response to the distractor, run 1, trial 11, female face, negative valence, alcoholic beverage distractor, nonmatch. Abbreviations:

SC: Skin Conductance

HR: Heart Rate

SpO2: pulse oximeter oxygen saturation

SCR: Skin Conductance Response

WAISIII: Wechsler Adult Intelligence Scale, Third Edition

VIQ: Verbal Intelligence Quotient

PIQ: Performance Intelligence Quotient

FSIQ: Full Scale Intelligence Quotient

WMSIII: Wechsler Memory Scale, Third Edition

Click here for additional data file.

Table S1 Characteristics of the research participants from whom skin conductance measurements were obtained

Means and standard deviations (SD) are displayed for age, education, measures of drinking history, the Wechsler Adult Intelligence Scale (WAIS) and the Wechsler Memory Scale (WMS). Besides amount and duration of drinking, these characteristics did not differ significantly between the ALC and NC groups.

Click here for additional data file.

Table S2 Characteristics of the research participants from whom heart rate measurements were obtained

Means and standard deviations (SD) are displayed for age, education, measures of drinking history, the Wechsler Adult Intelligence Scale (WAIS) and the Wechsler Memory Scale (WMS). Besides amount and duration of drinking, these characteristics did not differ significantly between the ALC and NC groups.

Click here for additional data file.

Burke Q. Rosen assisted with data processing and statistical analyses. Diane Merritt and Maria Valmas helped with patient recruitment and neuropsychological assessment.

Additional Information and Declarations

Competing Interests

Author Contributions

Human Ethics

Data Deposition

The authors declare there are no competing interests.

Kayle S. Sawyer conceived and designed the experiments, performed the experiments, analyzed the data, wrote the paper, prepared figures and/or tables, reviewed drafts of the paper.

Alan Poey performed the experiments, analyzed the data, reviewed drafts of the paper.

Susan Mosher Ruiz conceived and designed the experiments, performed the experiments, wrote the paper, prepared figures and/or tables, reviewed drafts of the paper.

Ksenija Marinkovic conceived and designed the experiments, analyzed the data, reviewed drafts of the paper.

Marlene Oscar-Berman conceived and designed the experiments, wrote the paper, reviewed drafts of the paper.

The following information was supplied relating to ethical approvals (i.e., approving body and any reference numbers):

This research was approved by the Institutional Review Boards of Boston University School of Medicine (H24686), VA Boston Healthcare System (1017+1018), and Massachusetts General Hospital (2000P001891).

The following information was supplied regarding the deposition of related data:

Figshare: http://dx.doi.org/10.6084/m9.figshare.1025792

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
