# Peer review of "Measures of skin conductance and heart rate in alcoholic men and women during memory performance"

_PeerJ, doi:10.7717/peerj.941_

## Round 0.1 · original submission · Major Revisions

· Academic Editor

Major Revisions

Dear Authors,Two peer reviewers have given very good suggestions to improve your manuscript. Please proceed to do the necessary revisions. We hope that the revised manuscript can be submitted soon so it can be re-reviewed.

·

Basic reporting

The present study describes a study investigating differences in physiological responses to emotional and alcohol stimuli between abstinent-alcoholic and non-alcoholic people. The experiment potentially has merit, however there were substantial limitations to the present manuscript that need to be addressed. Specifically:

1) The main purpose of the experiment was not clear to me. The paper should describe the purpose of looking at the SCR and HR measures. For example, are they thought to assess different mechanisms than the 'central nervous system' and behavioural deficits mentioned on line 19? Or are these measurements an alternative way of measuring these central nervous system deficits? If they are thought to be measuring different mechanisms of alcohol addiction, then the link between these mechanisms and alcohol addiction should be described. If it is the latter, then the research question should presumably be whether the HR and SCR measures are superior measures of these common mechanisms.

2) The introduction summaries past findings showing that alcoholic people show higher reactivity than non-alcoholic. However, the gap in the literature is not clear. What is this study adding to this basic knowledge?

Experimental design

3) It is not clear whether this is an exploratory study or whether it is testing specific hypotheses. This needs to be clearly stated, and if there were a-priori hypotheses than these need to be clearly stated.

4) Related, the purpose of the task design is not clear. Why is the memory task designed to pair emotional faces with alcohol/non-alcohol stimuli? Are the emotional expressions meant to prime responses to alcohol cues? Are the alcohol cues meant to interfere with the memory test? The function of the task design needs to be explained.

5) The problems mentioned on line 85 sound severe. ( MRI radio frequency interference, motion
induced artifact from the static magnetic field, poor skin responsivity with electrodes, other
motion artifact, etc.). This is a potentially fatal flaw in the methodology. Substantial reassurance that the remaining data is valid is required.

Validity of the findings

5) The analysis are not presented in sufficient detail.

6) The discussion does not integrate and form conclusions from the results.

7)The present findings are not linked back to the literature, rather the discussion raises new points seemingly unconnected to the present study.

Additional comments

Minor points:
Line 5: The introduction to the task at the end of the first paragraph is hard to follow, as there is no indication of the function of the alcohol/non-alcoholic distractors. When this component of the task is introduced, it would be good if it was accompanied by an explanation.

Line 31:“Electrodermal activity is an excellent indicator of overall arousal state, as it pertains to emotion, attention, and cognitive load in direct relationship to stimulus novelty, intensity, and significance” - This seems to general a statement. It needs to link back to an explanation of alcohol use and compared to other indicators of that dysfunction.

Line 36: it is not clear whether this section is proposing that HR is a measure of HPA axis and that SCR of limbic function (and can they consider AC as limbic)? If so this should be stated clearly. If it is just an association, and the two measures are might not be uniquely reflecting HPA and limbic systems then this paragraph doesn't seem necessary.

45: Maybe change wording here, it sounds like the 'however' in the last sentence is countering the rest of the paragraph. If the meta-analysis shows that there are differences overall, then it doesn't matter that individual studies did not find it. Make the meta-analysis sentence stronger, and put the 'while not all'on the negative findings. This paragraph needs to then indicate what is not known from these past studies that the present study can address.

56: this paragraph seems to be detailing alcohol priming studies. The link to the present study is not clear to me.

80: This is just a suggestion, but it would be easier to follow the participants section if it was in the reverse order. That is I would prefer the defininitons of the what constitutes a alcoholic [i.e., the dsm criteria] first, then second mention how that was determined [ie., structured interview] and finally a discussion of lost data.

84: These problems sound severe. I would appreciate some reassurance that the remaining data is valid.
Second, where is the MRI data. Was that all lost?

138: Mention the purpose of the neuropsych assessment. Was this just demographics, or part of the exploratory analysis?

Figure 1: picture could do without the left pointing arrow head

160: This is a clear description of the task, but not a explanation the task design. Why is it a memory task? How does the design of the task link to the processes being assessed? Why are faces used? Why are the alcohol cues there? Why are they presented in this manner? Are the emotional expressions meant to prime responses to alcohol cues? Are the alcohol cues meant to interfere with the memory test?

229: "Models were constructed for the encoded faces, distractor, and probe effects separately. The
within-subjects model was specified to include only the significant interactions and the
predictors of interest." This sentence is not clear to me. The models were constructed separately, sounds like you ran seperate analyses for each of those task components. However, you then present analyses of the interactions between components.

238: It is not clear why the encded faces are not reported. If they pretain to the hypotheses they should be reported. If it was exploratory then it should be reported. If they are irrelevant to the hypotheses then why are they assessed? It also looks like the emotional faces are in the following analyses, so it is not clear what has been not reported here.

255: The breakdown of the three way interaction should be accompanied with appropriate statistical tests (i.e., is the SCR for the Nonalc stim significantly lower than the SCR for the alc stim). If this is exploratory then the non-sig result are likely to be due to chance.

Discussion:
323: 324: where are these anlayses reported? The main effect doesn't seem to be in the results section.

335: 342: these paragraphs don't relate to the findings

Reviewer 2 ·

Basic reporting

No Comments

Experimental design

No Comments

Validity of the findings

No Comments

Additional comments

The paper would be strengthened by addressing the following minor points:
1. The terms SCL and SCR are defined as being different on page 2 but the authors seem to shift from one to the other as if they are interchangeable. There should be more clarity in use of these terms.
2. Table 1 includes length of sobriety for non-alcoholic controls. It is not clear what this means for controls since they were not alcoholics. This should be clarified.
3. Table 1 shows that mean duration of heavy drinking (greater than 21 drinks per week) was 0.3 years in non-alcoholic men. It is unclear why any subject who drank heavily was included in the non-alcoholic control group. This should be explained.
4. Alcoholic and non-alcoholic groups differed in education. Whether this could have affected the results warrants discussion.
5. In Figure 1, the asterisk after the encoded faces is shown but asterisks after the distractor and probe face are missing.

---

## Round 0.2 · Minor Revisions

· Academic Editor

Minor Revisions

Dear Authors,There are still revisions of the manuscript that needs to be done .Please read the comments of the peer reviewer carefully.The re-revised manuscript will be re-submitted again after your revised manuscript is resubmitted .

·

Basic reporting

The reporting was not suffient. There is need for greater clarity throughout the manuscript.

Experimental design

The experimental design is not linked to clear hypotheses, making it hard to interpret whether the design is relevant for the research question.

Validity of the findings

The findings are insufficiently interpretted.

Additional comments

I am still struggling to understand the basics of the paper; which makes it hard to know what needs to be clarified.

In the introduction, there are several theoretical points made, but these aren't brought together into a coherent set of hypotheses that are specific enough to test. An importantly notable absence here is that the hypotheses still don't reference the experimental design. By specific hypotheses I mean things such as: Men would show elevated SCR on X part of the task. This would be modified by the preceding Y part of the task.
If there were no specific hypotheses a priori, but a suspicion that there would be a difference somehow; then it should be framed as an exploratory study.

Method: I appreciate the reassurance about the quality of the data. I don't have the expertise to interpret the data presented to tell if it is actually ok.

Results: The results are still unclear to me. I think (judging by the reference to Figure 2), that the dependent variable being discussed in the paragraph begining on line 261 is the scr over the distractor. If this is the case it should be said up front. If not, clearly something needs to be clarified.
The statement that only the significant interactions where included is also confusing to me. Does this mean that the model was run, and then run again, only including the significant interactions? If so, then that seems like a data analytic approach that requires further justification.
Overall, as it is, the results are hard to decipher. Statements that explain the results relative to the hypotheses would make it substantially easier to interpret. For example, the introduction mentions that the study is looking at the effect of the distractors on the memory. I assume that the interaction starting on line 277 is relevant to this. A statement such as: “This result is [in]consistent with the hypothesis that the distractor would affect SCR ... etc.” would help the reader make sense of the data.

Discussion: There is similarly need for greater interpretation in the discussion. This will clearly depend on the hypotheses that are presented.
For example, the statement that :
“Thus, our results clearly demonstrated that the intensity of ANS responses differed for ALC and NC groups according to (a) the type of emotional stimulus (facial expressions and beverage cues), or (b) the temporal order in which the stimulus elements were presented”
is just a restatement of the results, as is the statement that there were variable effects. What is the interpretation of these differences? For example, does this mean that alcoholic people are more sensitive to all emotional stimuli, just to alc stimuli, that their memory is impaired? Or alternatively are the results not sufficiently clear to draw a conclusion? Are these differences consistent with other measures?

Overall the paper lacks narrative. There is data presented but it is not interpreted for the reader.

Reviewer 2 ·

Basic reporting

No comments

Experimental design

No comments

Validity of the findings

No comments

Additional comments

Appropriate changes have made in response to my previous review.

---

## Round 0.3 · Major Revisions

· Academic Editor

Major Revisions

Dear Authors,

Enclosed are comments from Reviewer 1 for your further action. Please do the address these comments and resubmit the manuscript again for re-review.

·

Basic reporting

The hypotheses are not clearly stated

Experimental design

Experimental design is not clearly related to the hypotheses and the results

Validity of the findings

It is unclear what findings were predicted a-priori

Additional comments

This revision is a substantial improvement on previous versions, and I commend the authors for their efforts.

However, the main comments I have made in the previous reviews remain unaddressed.
I am a bit confused about why; I'm not sure whether there is some miscommunication here. So I'll try to expand on my comments to a greater degree than in the previous reviews.

Firstly, there are no specific testable hypotheses in the introduction. The introduction currently states "We predicted that ALCs would have abnormal
physiological responses to the two different emotional characteristics (facial expressions and beverage cues), and that these responses would be different for men and women."
What is missing here is the statement of how the differences will be abnormal.
Would you expect a increase or decrease in responsivity? Would this vary according to task element? If so which task element?
For example, one obvious hypothesis that I would expect is that, relative to nonalc participants, alc participants would have greater reactivity to alc stimuli than to non-alc stimuli.
Similarly for each of your main findings it is not clear which findings were predicted and confirmed, and which were not. Each of the following could be a potential confirmed hypothesis:
"SCR results indicated lower sympathetic arousal levels in alcoholics"
"Our results indicated lower skin conductance reactivity for the ALC group than for the NC group in response to the encoded faces"
"We further identified that HR fell more for the ALC group than the NC group in response to the encoded faces"
"ALCs showed reduced SCRs to the distractors as a whole, there was not evidence for differential effects that were specific to the alcohol cue distractor images".
Prior to the results we should know whether each of these findings was expected. Alternatively, if there was a suspicion of abnormality, but none of these findings were specifically predicted in advance, then the paper should be labelled exploratory.
This would have important implications for assessing the likelihood that any particular sig finding could reflect a type 1 error.

The presentation of the results at the moment also reads like an exploratory study, in that only the significant results are discussed. Alternatively if there were specific hypotheses, I would expect presentation of the findings related to those hypotheses whether or not they were significant.
Indeed it seems strange that there is no reporting of the difference in reactivity to type of distractor stimuli in the SCR results, nor with the face stimuli.
But whether these results are important would depend on whether the hypothese specifically related to these findings.

On a related note, the relevance of the task design still alludes me. It looks like the task is designed to assess the effect of the distractor on the face memory, but this is hardly mentioned in the document. I would assume, from the design, that this effect and its relationship to the physiological measures is central to the study. The design implies the questions: Did the presence of the alcohol distractor affect the face recognition; did this vary by group; did this relate to the physiological measures? Indeed the quite sophisticated task design seems irrelevant to the current results section. The relevance of the design needs to be addressed.

---

## Round 0.4 · Major Revisions

· Academic Editor

Major Revisions

Dear Authors,
There are still issues that need to be resolved in the revised manuscript. Please revise it so as it can be re-reviewed in the next round.

·

Basic reporting

The basic reporting is sufficient, apart from a few minor details (detailed comments below)

Experimental design

The experiemental design is not linked to findings in the results (see comment 1 below)

Validity of the findings

The findings are generally presented well, apart from minor concerns ( comments 2, 3, 5 below)

Additional comments

Overall this is a marked improvement on the previous version, and I am comfortable that my main comments have been addressed.

However there still are some outstanding issues that I believe need to be addressed before publication. Firstly, the recent addition of the secondary goals in method section successfully addresses the reason for the experimental design (i.e., the section: “The secondary goal of the task was to examine the interaction of these factors in order to characterize the influence of the distractor on the memory for the faces.”). However, I was unable to see this goal addressed in the results section. I also think that it would be appropriate for this goal, and the specific testable hypotheses that come from this goal added to the introduction, and the relevant results to be added to the results section (i.e., was there a influence of the distractor type on the memory for the faces? And did this relate to the physiological measures ?)

Second, the start of the results states: “The results of the SCR and HR measures were in line with our predictions that [...] (b) greater reactivity to the alcoholic beverage stimuli than to the distractor stimuli unrelated to alcohol.” However, this doesn't seem to match the following detailed description of the results (e.g., “Group x Distractor interaction was not identified”, “Our hypothesis that differential HR responses to the distractor type would differ by group was not confirmed (F (2,78) = 0.72, p = 0.49))”, nor does it match the discussion.

Third, I would find it more helpful if the graphs matched the primary hypotheses. Specifically, show the results that are of most interest, including the hypothesis that the “greater reactivity to the alcoholic beverage stimuli than to the distractor stimuli unrelated to alcohol” by displaying the difference between the alcohol and non-alcohol distractors.

Fourth, the abstract should state the main hypotheses and findings (i.e., say both (a) relative to NC participants, the ALC participants would show reduced SCR and HR responses to the face stimuli, and (b) greater reactivity to the alcoholic beverage stimuli than to the distractor stimuli unrelated to alcohol)

Fifth, presumably it is significant that there was no interaction involving the face valence, this should be discussed in the discussion. At the moment the emotional face section of the discussion seems to solely focus on the main effect of group.

A final minor point: I would prefer it if the results that “Approached significance” had exact p values so the reader can see how much it approached significance, rather than using p < .1.

---

## Round 0.5 · accepted · Accept

· Academic Editor

Accept

Congratulations,The manuscript is accepted for publication and will undergo galley proof preprocessing .

·

Basic reporting

The basic reporting is sufficient

Experimental design

Description of the design is sufficient

Validity of the findings

Findings reported are appropriate

Additional comments

The authors have sufficiently addressed my concerns.

Minor comment: On P10 row 195 'its' is unclear. It is not clear whether the 'its' is referring to Face Emotion.